# Germline and somatic albinism variants in amelanotic/hypomelanotic melanoma: Increased carriage of *TYR* and *OCA2* variants

Jenna E. Rayner[1], David L. Duffy[1,2], Darren J. Smit[1], Kasturee Jagirdar[1], Katie J. Lee[1], Brian De'Ambrosis[3,4,5], B. Mark Smithers[6], Erin K. McMeniman[1,3], Aideen M. McInerney-Leo[1], Helmut Schaider[1], Mitchell S. Stark[1], H. Peter Soyer[1,3], Richard A. Sturm[1]*

1 Dermatology Research Centre, The University of Queensland Diamantina Institute, The University of Queensland, Brisbane, Qld, Australia, 2 QIMR Berghofer Medical Research Institute, Brisbane, Qld, Australia, 3 Dermatology Department, Princess Alexandra Hospital, Brisbane, Qld, Australia, 4 Faculty of Medicine, The University of Queensland, Brisbane, Qld, Australia, 5 South East Dermatology, Annerley, Brisbane, Qld, Australia, 6 Queensland Melanoma Project, School of Medicine, The University of Queensland, Brisbane, Qld, Australia

* r.sturm@uq.edu.au

**Data Availability Statement:** All relevant data are within the manuscript and its Supporting Information files.

## Abstract

Amelanotic/hypomelanotic melanoma is a clinicopathologic subtype with absent or minimal melanin. This study assessed previously reported coding variants in albinism genes (*TYR*, *OCA2*, *TYRP1*, *SLC45A2*, *SLC24A5*, *LRMDA)* and common intronic, regulatory variants of *OCA2* in individuals with amelanotic/hypomelanotic melanoma, pigmented melanoma cases and controls. Exome sequencing was available for 28 individuals with amelanotic/hypomelanotic melanoma and 303 individuals with pigmented melanoma, which were compared to whole exome data from 1144 Australian controls. Microarray genotyping was available for a further 17 amelanotic/hypomelanotic melanoma, 86 pigmented melanoma, 147 melanoma cases (pigmentation unknown) and 652 unaffected controls. Rare deleterious variants in *TYR*/OCA1 were more common in amelanotic/hypomelanotic melanoma cases than pigmented melanoma cases (set mixed model association tests $P = 0.0088$). The *OCA2* hypomorphic allele p.V443I was more common in melanoma cases (1.8%) than controls (1.0%, $X^2$ $P = 0.02$), and more so in amelanotic/hypomelanotic melanoma (4.4%, $X^2$ $P = 0.007$). No amelanotic/hypomelanotic melanoma cases carried an eye and skin darkening haplotype of *OCA2* (including rs7174027), present in 7.1% of pigmented melanoma cases ($P = 0.0005$) and 9.4% controls. Variants in *TYR* and *OCA2* may play a role in amelanotic/hypomelanotic melanoma susceptibility. We suggest that somatic loss of function at these loci could contribute to the loss of tumor pigmentation, consistent with this we found a higher rate of somatic mutation in *TYR*/OCA2 in amelanotic/hypomelanotic melanoma vs pigmented melanoma samples (28.6% vs 3.0%; $P = 0.021$) from The Cancer Genome Atlas Skin Cutaneous Melanoma collection.

**Funding:** This work was supported by the National Health and Medical Research Council (NHMRC; https://www.nhmrc.gov.au/) Project (APP1062935), the Centre of Research Excellence for the Study of Naevi (APP1099021), an NHMRC Practitioner Fellowship (APP1137127) to HPS, and NHMRC Fellowships (APP1106491 and APP1158111) to MSS and AML. Funding for WES was provided by Queensland Genomics, Queensland Government, Round 1 Demonstration Project grant (https://queenslandgenomics.org/). The funders had no role in study design, data collection and analysis, decision to publish, or preparation of the manuscript.

**Competing interests:** I have read the journal's policy and the authors of this manuscript have the following competing interests: HPS is a shareholder of MoleMap NZ Limited and e-derm consult GmbH, and undertakes regular teledermatological reporting for both companies. HPS is a Medical Consultant for Canfield Scientific Inc. and MetaOptima Technology Inc., a Medical Advisor for First Derm, and has a Medical Advisory Board Appointment with MoleMap NZ Limited. This does not alter our adherence to PLOS ONE policies on sharing data and materials. The other authors have no conflicts of interest to declare.

## Introduction

Albinism is a group of severe genetic disorders characterized by reduced or absent biosynthesis of melanin pigment in melanocytes of the skin, hair follicles and eyes [1]. It has been classically subdivided into three groups: oculocutaneous (OCA), ocular (OA), and syndromic albinism. Most forms of OCA are characterized by features including nystagmus, foveal hypoplasia, iris transillumination, and photophobia [2]. In European populations ~1/17,000 people are affected, but there may be under diagnosis in light skinned Europeans [3] compared with ethnic groups of darker pigmentation.

All seven known types of non-syndromic OCA are autosomal recessive in inheritance. OCA1A is characterized by the complete absence of melanin production throughout life due to the absence of *TYR* gene activity. The OCA1B subtype is a milder form that displays some pigment accumulation over time with a low degree of retained tyrosinase (TYR) enzyme activity [1]. Other albinism classes are: OCA2 caused by deletion or loss of activity of the melanosomal P-protein (*OCA2* gene); OCA3 attributable to mutations in *TYRP1*; OCA4 due to loss of *SLC45A2*; and OCA5 which is linked to chromosome 4q24, with the responsible gene yet to be characterized. More recently OCA6 and OCA7 have been identified with mutations in *SLC24A5* and the *LRMDA* genes respectively [1]. Of those who receive a molecular diagnosis, OCA1 and OCA2 account for 42% and 28% of cases respectively [3].

Common variants in several of these genes are associated with normal as well as pathological variation in human pigmentation [4–7]. An important example is *OCA2* [MIM 611409], where protein coding or regulatory variants alter the expression or function of the P-protein, which assists trafficking and processing of the TYR protein [8]. Deletion of the region encompassing the *OCA2* gene on chromosome 15, as observed in Prader-Willi and Angelman syndromes, is associated with hypopigmentation of the skin, hair, and eyes [9], and extra copies of this chromosomal region result in generalized hyperpigmentation of the skin [10, 11]. The greatest contributor to normal variation in eye color is a common intronic single nucleotide polymorphism, rs12913832 in *OCA2* [12, 13], that also affects skin color, but has a disproportionately small magnitude of effect on melanoma risk [14]. Importantly, heterozygote genotypes exhibit intermediate effects in these situations.

An amelanotic/hypomelanotic melanoma (AHM) is a tumor subtype with no or little melanin [15, 16]. AHM comprises 2% to 8% of all melanoma cases, and is more common in those who are over the age of 50 years, have red hair and/or fair skin type [17]. We have previously shown the common *MC1R* and *TYR* variants associated with fair skin occur at increased frequency in AHM [18]. Therefore, it is reasonable to hypothesize that rarer albinism-associated variants should also be more frequent in AHM cases as compared to pigmented melanoma (PM). However, we also note that in patients with multiple primary melanomas (MPM), individual tumors can be more or less pigmented than their predecessors.

This study aimed to compare the rates of rare alleles in albinism genes in patients with AHM, PM only, or with no melanoma history and found differences in rates of *TYR*/OCA1 and *OCA2* variants.

## Materials and methods

### Ethics approval and data availability

This study was approved by the Human Research Ethics Committee of Princess Alexandra Hospital (approval HREC/09/QPAH/162) and The University of Queensland (approval #2009001590) and conducted in accordance with the Declaration of Helsinki. Participants provided written informed consent and parental/guardian consent was also provided for those

under 18 years. Three variant call files for AHM, PM and control Brisbane Naevus Morphology Study (BNMS) participants respectively for all alleles seen in the regions encompassing the 10 genes described in this report are provided (S1–S3 Files). This combines the 1233 case-control participants analysed by Illumina SNP genotyping with those seen in whole exome sequencing (WES) of 383 melanoma cases (S1 Fig). A separate Excel file containing ANNOVAR annotated variants of these 10 genes is also provided (S4 File), and specific *in silico* analysis of *TYR* and *OCA2* variants of unknown significance are shown in S5 File.

## Data and pathology collection

A research assistant collected histopathological reports of participants' melanomas from pathology providers or the Queensland Cancer Registry database. Melanomas were included if described as "amelanotic" on the histopathology summary, or as hypomelanotic melanomas when both 1) the clinician queried hypomelanotic pathology on the request form e.g.? amelanotic/?BCC/?fibroma/?IEC/?cyst AND 2) the pathology specimen was described macroscopically by the processing technician as cream, pink, pale or flesh colored.

## Biospecimen collection and genotyping

Genomic DNA was extracted from 2ml of saliva collected using an Oragene-DNA self-collection kit (DNA Genotek, Ottawa, ON, Canada). A minimum of 2.5ug of DNA was processed at the University of Queensland Centre for Clinical Genomics at the Translational Research Institute on an Illumina Infinium HumanCoreExome-24 Microarray [19–21]; 426 of 436 case samples and all 692 controls were of sufficient quality for genotyping.

## Whole exome sequencing and data analysis

A total of 383 pathology report confirmed cutaneous melanoma individuals, identified as high risk with respect to age, melanoma and naevus counts, were selected from the 1266 BNMS participants [22]. Exome library preparation (SureSelect V5+UTR; Agilent, CA, USA) on 1ug saliva-derived DNA and whole exome sequencing was performed by a fee-for-service provider, the Australian Genome Research Facility using the Illumina platforms HiSeq 2500 (n = 1–24) and NovaSeq 6000 (n = 25–383) to > 60x targeted depth coverage. Mapping and annotation was performed as previously described [23] or as detailed herein. Briefly, 125bp paired end reads were mapped to the UCSC hg-19 reference sequence using BWA [24] alignment software optimized to run on Edico Genome's Dragen FPGA (Dragen v01.011.222.02.04.02.34804). The alignments were further filtered using Picard tools (https://github.com/broadinstitute/picard), Samtools [25] and GATK toolkit (https://software.broadinstitute.org/gatk/) using standard workflows. All variants were called using mpileup [24] and VarScan2 [26]. To reduce the number of false positive variant calls, a minimum threshold of 25% variant frequency was set in VarScan2. All variants were subsequently annotated using ANNOVAR [27] including dbSNP138, 1000 Genomes Project and Exome Aggregation Consortium (ExAC) databases. Prediction of variant effect was determined using ANNOVAR [27] version dbNSFP [28], including *in silico* predictors SIFT [29], Polyphen2 [30], LRT [31] and Mutation Taster [32]. We used external controls from the Medical Genome Reference Bank (MGRB [33]), which is comprised of whole genome sequencing data and phenotypic information from (up to) 4,000 healthy Australians over 70 years of age. MGRB participants, consented through contributing studies, 45 and Up (Sax Institute, Sydney), and the Aspirin in Reducing Events in the Elderly clinical trial (Monash University, Melbourne), are free from cardiovascular disease, degenerative neurological disorders and of a history of cancer at the time of consent into the study.

We sourced a secondary data set of germline and somatic DNA sequence from the NIH National Cancer Institute Genomic Data Commons Data Portal, The Cancer Genome Atlas (TCGA) program consisting of whole genome sequence from 10,389 cancer patients which included 470 melanoma cases [34].

### Variant classification and analysis

Rare germline variant analysis was undertaken using a candidate gene panel (S1 Table–S3 Table) which focused primarily on genes relating to pigmentation, including albinism genes. The candidate gene variants were subjected to additional filtering including: 1) Phred variant base quality of 30;2) a total read depth coverage of $\geq$10;3) allele frequency $\leq$ 1% in ExAC/gnomAD database Non-Finnish European population; 4) variants considered as protein-altering (missense, nonsense, and stoploss as well as frameshift deletion/insertion and non-frameshift deletion/insertion) present in a coding exon; and 5) exonic and intronic splice-site regions. Variants were filtered further to include only 'deleterious' mutations, which were predicted to be damaging in at least one *in silico* prediction tool. In our dataset, PolyPhen2 (HVAR) and MutationTaster provided the most consistent correlation with known naevi and melanoma-related genes. All loss of function variants (stopgain and frameshift) were considered deleterious.

### Statistical analysis

Statistical analysis was conducted using Sib-pair 1.0.0 computer program [35] and the R statistical computing environment [36]. Individual SNP association tests were carried out using a bias-reduced logistic regression *logistf* R package [37]. Genotype data was cleaned by removing individual genotype GC scores <0.6, and all genotypes at SNPs where Hardy-Weinberg disequilibrium *P*-value $<10^{-5}$, or where no homozygotes were observed. Allele frequency of each variant in each subgroup was compared using contingency $X^2$ tests. All P-values are unadjusted for multiple testing, but should be interpreted as nested within the gene based tests. Set Mixed Model Association Tests (SMMAT, [38]) were conducted to combine evidence from rare variants in the same genes or pathways, allowing for some relatedness in our case samples. The SMMAT weighted gene based test in the GENESIS package was calculated first using only variants available from the WES. This correctly adjusts for the presence of multiple variants contributing to the test. We subsequently expanded our analysis to include our microarray genotyped BNMS controls to increase statistical power and include controls that are even more closely matched than the MGRB collection.

## Results

### Analysis of albinism gene alleles in PM and AHM cases

A total of 581 melanoma case and 652 control participants drawn from the Brisbane Naevus Morphology study (BNMS) were previously genotyped using a high density Illumina SNP array panel [22]. In earlier work we compared the phenotypic and candidate genotypic characteristics of 45 subjects with AHM, 25 of these with MPM, and 389 with pigmented melanoma cases, 203 with MPM [18]. In the present study, we performed whole exome sequencing (WES) to detect rare, coding germline variants in a subgroup of 383 melanoma case patients (28 AHM, 303 PM, 52 unknown pigmentation status, S1 Fig).

We first looked for coding region polymorphisms in the six reported OCA genes: *TYR*, *OCA2*, *TYRP1*, *SLC45A2*, *SLC24A5* and *LRMDA* (S1 Table and S2 Table). The frequency of all corresponding alleles present in the gnomAD database [39] and in 1144 genomes from the

MGRB [33] are also listed. The latter also allows statistical testing of mutation burden versus an ethnically closely-matched control sample.

A total of 20 coding region variants (S1 Table, gnomAD minor allele frequency (MAF)) were seen for the *TYR* gene with the two common alleles p.S192Y (36.4%) and p.R402Q (27.3%) having been previously reported in this dataset [18]. These findings were validated in this study with the common p.402Q allele occurring at a higher frequency in the AHM versus PM group (42.2% vs 32.6%; $P = 0.07$; $P = 0.015$ when compared to combined MGRB + BNMS controls). In addition to these common variants, there were 18 rare coding variants with <1% frequency in gnomAD [39] or MGRB control collection, 13 of which have been previously reported to cause OCA1 [3] (Albinism database http://www.ifpcs.org/albinism/index.html). The remaining five were predicted to be deleterious using *in silico* prediction tools. Three of these *TYR* rare variants (p.A23T, p.T373K and p.P460L) were found in the AHM population in comparison to six (p.R217Q, p.V275F, p.R299H, p.N371T, p.T373K, p.P460L) in the larger PM group. Overall, rare *TYR* variants were at a MAF of 4.67% in AHM cases compared to 1.76% in PM cases, and 1.14% in MGRB controls (Table 1 and S1 Table). In considering the frequency of individual rare *TYR* variants in different subtypes of melanoma, the p.A23T

**Table 1.  *TYR* and *OCA2* deleterious alleles in pigmented melanoma and amelanotic/hypomelanotic melanoma patients.**

| Gene/OCA | Controls | | Melanoma Cases | | | SMMAT Tests (*P*-value) | | |
|---|---|---|---|---|---|---|---|---|
| | MGRB Control N (combined MAF %) Total = 1144 WES | BNMS Control N (combined MAF %) Total = 652 [a] | PM N (combined MAF%) Total = 389 [a] (WES = 303) | AHM N (combined MAF %) Total = 45 [a] (WES = 28) | Total Melanoma Cases N (combined MAF %) Total = 581 [a] (WES = 383) | Total Melanoma Cases vs MGRB + BNMS Controls | AHM vs PM Cases | AHM Case vs MGRB + BNMS Controls |
| ***TYR*/OCA1** | | | | | | | | |
| **Total number of *TYR* alleles (MAF<1%) observed in each subgroup** | 26 (1.14) | 4 (0.31) | 12 (1.76) | 3 (4.67) | 17 (2.22) | 0.065 | **0.0088** | **1.7x10⁻¹⁴** |
| ***OCA2*/OCA2** | | | | | | | | |
| **Total number of *OCA2* alleles (MAF<1%) observed in each subgroup [b]** | 21 (0.90) | 6 (0.45) | 6 (0.95) | 0 (0.00) | 7 (0.91) | 0.61 | 0.81 | 0.84 |
| ***TYR*/OCA1 + *OCA2*/OCA2** | | | | | | | | |
| **Total number of *TYR* and *OCA2* alleles (MAF<1%) observed in each subgroup [b]** | 47 (2.04) | 10 (0.76) | 18 (2.71) | 3 (4.67) | 24 (3.13) | **0.008** | 0.90 | 0.095 |
| ***OCA2* p.V443I variant** | Controls | | Melanoma Cases | | | $X^2$ Statistical Tests (*P*-value) | | |
| rs121918166*C/T p.V443I | 25 (1.09) | 11 (0.84) | 12 (1.57) | 4 (4.44) | 21 (1.81) | **0.0413** | 0.0935 | **0.0097** |

[a] Illumina Core Exome genotyping

[b] Does not include *OCA2* rs121918166*C/T p.V443I

AHM, amelanotic/hypomelanotic melanoma

BNMS, Brisbane Naevus Morphology Study

MAF, minor allele frequency

MGRB, Medical Genome Reference Bank

N, number

PM, pigmented melanoma

WES, whole exome sequencing

variant showed the largest difference, occurring at a higher frequency in AHM vs PM ($X^2$ unadjusted $P = 0.008$). The p.A23T and p.T373K variants were more common in AHM cases as compared to controls ($X^2$ unadjusted $P = 0.006$ and $X^2$ $P = 0.02$ respectively). Analyzing the total burden of *TYR* rare variants together, using a SMMAT test [38], gave a significant difference for AHM vs PM (SMMAT $P = 0.0088$) and AHM vs controls (SMMAT $P = 1.7 \times 10^{-14}$). The *TYR* common variant double haplotype p.192Y-p.402Q [40–42], though at slightly higher frequency in AHM than PM cases (2.8% vs 2.1%), was not significantly different.

There were 18 coding variants in *OCA2* (S1 Table, gnomAD MAF), two of which, p.R305W (5.05%) and p.R419Q (6.5%), have been described previously as being common [12], with a third variant p.V443I at 1.09% in MGRB controls (Table 1 and S1 Table). Seven of the remaining 15 rare variants (<1%) have been reported as causative for OCA2 albinism [3, 43] (excluding p. R266W which has been reported as common in African populations) and seven were *in silico* predicted to be deleterious. None of these rare alleles were seen in the AHM group, but there were six seen each once in the PM group (p.A55fs, p.P211L, p.I370T, p.N489D, p.F685fs, p.L734R). When any BNMS case was compared with controls (Table 1) there was no significant difference in *OCA2* combined rare alleles (SMMAT $P = 0.61$). However, analyzing each variant separately revealed that the p.V443I variant occurred at higher frequency in AHM cases (4.44%) compared to PM cases (1.57%), BNMS controls (0.84%), or the MGRB (1.09%) (combined control comparison $X^2$ $P = 0.01$), with comparison of "any melanoma" nominally significant ($X^2$ $P = 0.04$).

Combining the *TYR* and *OCA2* rare alleles (excluding p.V443I due to MAF >1% in MGRB controls) together for statistical analysis (Table 1), there was a higher frequency in AHM than in PM cases (4.67% vs 2.71%), but this did not achieve statistical significance. However, compared to combined controls (3.13% BNMS/2.04% MGRB), there was a highly significant greater occurrence of *TYR+OCA2* deleterious alleles in total melanoma cases (SMMAT $P = 0.008$), but not AHM (SMMAT $P = 0.90$), nor in comparison between AHM and PM (SMMAT $P = 0.095$).

There were no rare variants in *TYRP1* (OCA3), *SLC45A2* (OCA4), *SLC24A5* (OCA6) or *LRMDA* (OCA7) in our AHM population. There was also no rare variants for *SLC24A5* (OCA6) in the melanoma samples, and no significant difference in frequencies between cases and controls for the four *TYRP1*, five *SLC45A2* and one *LRMDA* rare alleles that were observed in the study. The frequency of rare variants in these genes in the control populations are included in S2 Table.

The odds ratios (OR) were then evaluated for *TYR* rare variant frequencies and the *OCA2* p.V443I allele frequency. These were higher in melanoma cases compared to BNMS controls, with carriers having an OR of 3.28 (95% confidence interval (95%CI) 1.23–8.77) and 1.86 (95%CI 1.03–3.37) respectively (Table 2). Upon comparing AHM cases to BNMS controls, higher OR for both *TYR* at 14.52 (95%CI 2.87–31.58) and *OCA2* p.V443I at 5.17 (95%CI 1.80–14.85) were obtained. In each of these analyses, the combined rare variants of *TYR* were of higher penetrance than *OCA2* p.V443I alone. In considering *TYR* combined rare variants +*OCA2* p.V443I together, there were 2 cases with double heterozygosity for PM and one for AHM cases, with none in the BNMS controls. These small numbers resulted in very high OR but with broad confidence intervals, and a larger data set will be required to confirm the apparent synergistic penetrance of combined genotypes at *TYR* and *OCA2* p.V443I.

## Analysis of *HERC2/OCA2* blue eye color associated polymorphisms in AHM

Two haplotype regions of the *HERC2/OCA2* locus have been recognized to be associated with blue eye color and lighter skin [13, 44, 45]. Both of these regions were examined for association

**Table 2.** *TYR* variants and *OCA2* p.V443I in control vs any melanoma case and amelanotic/hypomelanotic melanoma patients.

| Gene | Controls versus any melanoma cases | | | | | Controls versus AHM cases | | | | |
|---|---|---|---|---|---|---|---|---|---|---|
| | Total N | MGRB Controls | BNMS Any Cases | Odds Ratio (95% CI) | Trend $X^2$ (P-value) | Total N | MGRB Controls | BNMS AHM cases | Odds Ratio (95% CI) | Trend $X^2$ (P-value) |
| ***TYR*** | | | | | | | | | | |
| 0/0 [a] | 1698 | 1138 (67.0) | 560 (33.0) | 1 | $5 \times 10^{-4}$ | 1833 | 1791 (97.7) | 42 (2.3) | 1 | $8 \times 10^{-5}$ |
| 0/1 | 16 | 6 (37.5) | 10 (62.5) | **3.28 (1.23–8.77)** | | 18 | 15 (83.3) | 3 (16.7) | **14.52 (2.87–31.58)** | |
| Total N (%) | 1714 (100) | 1144 (66.7) | 570 (33.3) | | | 1851 (100) | 1806 (97.6) | 45 (2.4) | | |
| ***OCA2* p.V443I** rs121918166*C/T | | | | | | | | | | |
| C/C | 1670 | 1120 (67.1) | 550 (32.9) | 1 | 0.03 | 1161 | 1120 (96.5) | 41 (3.5) | 1 | 0.011 |
| C/T | 44 | 23 (52.3) | 21 (47.7) | **1.86 (1.03–3.37)** | | 27 | 23 (85.2) | 4 (14.8) | **5.17 (1.80–14.85)** | |
| T/T | 1 | 1 (100) | 0 (0) | 0.68 (0.03–16.68) | | 1 | 1 (100) | 0 (0) | 9.0 (0.36–224.27) | |
| Total N (%) | 1715 (100) | 1144 (66.7) | 571 (33.3) | | | 1189 (100) | 1144 (96.2) | 45 (3.8) | | |
| ***TYR* + *OCA2* p.V443I** rs121918166*C/T | | | | | | | | | | |
| 0/0, C/C | 2290 | 1756 (76.7) | 534 (23.3) | 1 | $3 \times 10^{-6}$ | 1795 | 1756 (97.8) | 39 (2.1) | 1 | $1 \times 10^{-6}$ |
| 0/1, C/C | 31 | 15 (48.4) | 16 (51.6) | **3.50 (1.74–7.04)** | | 17 | 15 (88.2) | 2 (11.8) | **7.17 (1.82–28.30)** | |
| 0/0, C/T | 53 | 34 (64.2) | 19 (35.8) | **1.86 (1.06–3.26)** | | 37 | 34 (91.9) | 3 (8.1) | **4.5 (1.44–14.16)** | |
| 0/0, T/T | 1 | 1(100) | 0 (0) | 1.10 (0.4–26.93) | | 1 | 1 (100) | 0 (0) | 14.82 (0.59–369.57) | |
| 0/1, C/T | 2 | 0 (0) | 2 (100) | **16.43 (0.79–342.80)** | | 1 | 0 (0) | 1 (100) | **133.41 (5.35–3326.09)** | |
| 0/1, T/T | - | - (-) | - (-) | - | | - | - (-) | - (-) | - | |
| Total N (%) | 2377 (100) | 1806 (76) | 571 (24) | | | 1851 (100) | 1806 (97.6) | 45 (2.4) | | |

[a] WT allele indicated as 0, any rare variant allele as 1 as listed in S1 Table.

AHM, amelanotic/hypomelanotic melanoma

BNMS, Brisbane Naevus Morphology Study

MGRB, Medical Genome Reference Bank

N, number

with the AHM phenotype. Notably, the most significantly associated SNP for blue eye color, rs12913832*G/A (intron 86 of *HERC2*) (Fig 1), was not statistically associated with AHM (Tables 3 and 4).

The three-SNP haplotype (rs7495174, rs4778241, rs4778138) of the first intron of *OCA2*, predictive of eye color [12], was expanded to allow a larger region to be analyzed. The haplotype (Fig 1, spanning the *OCA2* 5'UTR to intron 2 region) listed by position in Table 3 showed a significant difference in allele frequency between the AHM cohort and PM case groups. Most significantly, the rs7174027*A SNP flanking the *OCA2* gene promoter region/transcription start site was absent in AHM patients, compared to a frequency of 7.1% in the PM group ($X^2$ unadjusted $P = 0.0005$) and 9.43% frequency in the BNMS control group ($X^2$ unadjusted $P = 8 \times 10^{-5}$). Three other SNPs, rs1470608*T, rs4778138*G and rs7495174*G, also showed a significant difference in comparing the AHM to the PM case group or BNMS controls. Rs4778138*G and rs7495174*G have previously been associated with blue eye color [12, 13].

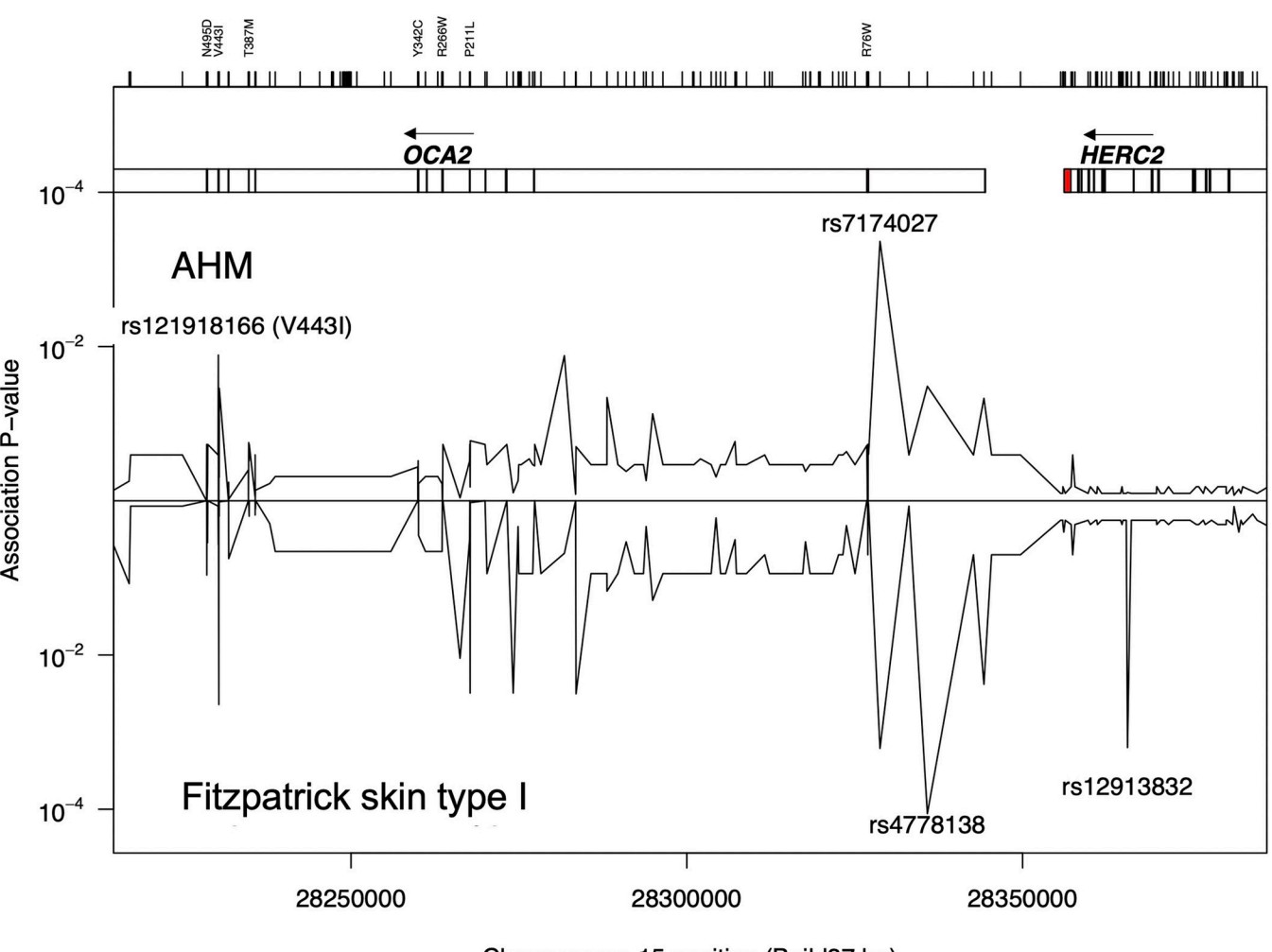

**Fig 1. Association $X^2$ P-values of the *HERC2/OCA2* gene promoter region and coding variants with amelanotic/hypomelanotic melanoma (upper) and skin phototype (lower, reading downwards).** The position of the peak SNPs are indicated above or below the plots. The *OCA2* and *HERC2* transcription units are boxed with exons are indicated by black bars and intronic regions in white. Coding variants and chromosome position are at the top and bottom of the plot respectively.

### Interaction of *HERC2/OCA2* rs12913832 and rs7174027 in blue eye color, sun sensitivity and AHM

Following the significant *OCA2* haplotype findings above, we analyzed the interaction of *HERC2/OCA2* rs12913832*G/A and rs7174027*G/A SNPs in blue eye color, sun sensitivity and AHM (Table 4, Fig 1). The *HERC2* rs12913832*A allele was further confirmed to be significantly associated with blue eye color ($P<2.2x10^{-16}$; OR 25.32 (18.16–35.92)). The rs7174027*A allele was most significantly associated with AHM ($P = 0.0002$; OR 19.67 (2.76.16–2495.52)) and to a lesser degree with blue eye color ($P = 0.0047$; OR 2.17 (1.33–3.54)). Both SNPs were significantly associated with sun sensitivity, rs12913832*A ($P = 0.0018$) and rs7174027*A ($P = 0.0036$) and with Fitzpatrick skin type I (OR 1.29 (1.03–1.61) and 1.60 (1.13–2.29) respectively). Despite the association of individual SNPs, there was no significant interaction in a joint model for blue eye color, sun sensitivity or AHM (Table 4).

**Table 3.** *HERC2/OCA2* gene non-coding allele/haplotype in pigmented melanoma and amelanotic/hypomelanotic patients.

| HERC2/OCA2 | Chr: Position (GRCh37) | HGVS Transcript Variant | Controls | | Melanoma Cases | | | X² P-value[e] | |
|---|---|---|---|---|---|---|---|---|---|
| | | | gnomAD [a] (MAF%) | BNMS Controls N (MAF %) Total = 652[d] | BNMS Cases N (MAF %) Total = 581[d] | PM N (MAF%) Total = 389 [d] | AHM N (MAF %) Total = 45 [d] | AHM vs PM Cases | AHM Case-Control |
| rs16950821*G/A | 15: 28283507 | NM_000275.2: c.228-6198C>T | A (10.17) | 128 (9.82) | 106 (9.12) | 72 (9.47) | 5 (5.56) | 0.177 | 0.127 |
| rs1470608*G/T | 15: 28288121 | NM_000275.2: c.228-10812C>A | T (13.64) | 197 (15.11) | 149 (12.82) | 104 (13.65) | 6 (6.67) | **0.043** | **0.014** |
| rs7174027*G/A | 15: 28328765 | NM_000275.2:c.-21-1724C>T | A (9.36) | 123 (9.43) | 78 (6.71) | 54 (7.1) | 0 (0.0) | **5x10⁻³** | **8x10⁻⁵** |
| [b,c] rs4778138*A/G | 15: 28335820 | NM_000275.2:c.-22+8550T>C | G (12.31) | 156 (11.96) | 114 (9.81) | 79 (10.37) | 4 (4.44) | **0.043** | **0.022** |
| rs7495174*A/G | 15: 28344238 | NM_000275.2:c.-22+132T>C | G (5.3) | 86 (6.6) | 46 (3.96) | 35 (4.59) | 1 (1.11) | 0.073 | **0.012** |
| rs12913832*G/A | 15:28365618 | NM_004667.5: c.13272+874T>C | A (19.6) | 304 (23.1) | 239 (20.57) | 161 (21.13) | 19 (21.11) | 0.996 | 0.71 |

[a] Lek et al., 2016 [39] http://exac.broadinstitute.org. European Non-Finish, listing the minor allele frequency.

[b] Previously rs6497268

[c] eQTL Zhang et al., 2018 [46]

[d] Illumina Core Exome genotyping

[e] Bonferroni corrected (two phenotypes x 6 variants) critical $P = 0.004$ equivalent to a table wide $\alpha = 0.05$

AHM, amelanotic/hypomelanotic melanoma

BNMS, Brisbane Naevus Morphology Study

GRCh37, Genome Reference Consortium Human Build 37

MAF, minor allele frequency

N, number

PM, pigmented melanoma

**Table 4.** Interaction of *HERC2/OCA2* rs12913832 and rs7174027 in blue eye colour, sun sensitivity and amelanotic/hypomelanotic melanoma.

| SNP | Total BNMS Cohort (N)[a] | Pr(>Chi)[c] Blue eye colour | Odds Ratio (95%CI) Blue eye colour |
|---|---|---|---|
| rs12913832 | 1228 | **< 2.2x10⁻¹⁶** *** | 25.32 (18.16–35.92) |
| rs7174027 | 1226 | **0.0047** ** | 2.17 (1.33–3.54) |
| rs12913832:rs7174027 | 1222 | 0.1094 | |
| | Total BNMS cohort (N)[a] | Pr(>Chi)[c] sun sensitivity | Odds Ratio (95%CI) Fitzpatrick Skin Type I |
| rs12913832 | 1231 | **0.0018** ** | 1.29 (1.03–1.61) |
| rs7174027 | 1229 | **0.0036** ** | 1.60 (1.13–2.29) |
| rs12913832:rs7174027 | 1225 | 0.7161 | |
| | Melanoma cases (N)[b] | Pr(>Chi)[c] AHM vs controls | Odds Ratio (95%CI) AHM |
| rs12913832 | 702 | 0.8721 | 0.79 (0.49–1.33) |
| rs7174027 | 700 | **0.0002** *** | 19.67 (2.76–2495.52) |
| rs12913832:rs7174027 | 697 | 1.0 | |

[a] Including cases and controls

[b] Pigmented melanoma and amelanotic/hypomelanotic melanoma (AHM) cases confirmed on pathology reports.

[c] Pr(>Chi), The probability that a particular Chi-Square test statistic is as extreme as, or more so, than what has been observed under the null hypothesis.

** $P<0.05$

*** $P<0.001$

AHM, amelanotic/hypomelanotic melanoma

BNMS, Brisbane Naevus Morphology Study

N, number

SNP, single nucleotide polymorphism

### Analysis of other human pigmentation gene variant alleles in AHM

For comparison with the albinism related genes, four other human pigmentation genes regulating melanogenesis, including *KITLG*, *POMC*, *SLC24A4* and *TPCN2*, were selected to study for any relationship to AHM (S3 Table). Although there were differences in frequencies in some of the common and rare alleles between AHM and PM cases, none of these alleles when combined, reached statistical significance for association with AHM or in comparison to the frequencies found in controls. None of the 45 AHM patients carried a *CDKN2A* gene mutation, with only one carrying the *MITF* E318K allele as previously described [21].

### Germline and somatic mutations of the *MC1R* and albinism genes in the TCGA SKCM collection

Data from 10,389 adult cancers from the Cancer Genome Atlas (TCGA) dataset [47] were first used to compare the common [4] and rare [48] *MC1R* variant allele frequencies in PM and AHM samples in the 470 SKCM (Skin Cutaneous Melanoma) collection. There were 7 AHM recognized in the documentation describing this dataset, with the remaining 463 considered as PM (Table 5 and S4 Table). Previously we [18] and others [49, 50] have reported a higher frequency of the highly penetrant *MC1R R* allele in AHM compared with PM patients. Although there were no R/R genotypes seen in the AHM samples they all carried either an *r* or *R* variant and no WT/WT, compared with a frequency of 19.7% of PM and 37.9% in other cancers. In comparing AHM and PM allele frequencies, WT was under represented (21.4 vs 41.9%) and both *r* (42.9 vs 31.1%) and *R* (35.7 vs 27.0%) were over represented. Thus the SKCM AHM samples are consistent with having a higher frequency of *MC1R* variants compared to PM.

The 7 AHM did not carry any deleterious germline deleterious or albinism alleles, nor the *OCA2* p.V443I polymorphism. However, upon examination of the sequencing data matched AHM tumor for somatic mutations or copy number variation (CNV) changes, four of the albinism genes were found to have deleterious mutations present in one or more samples (Table 5). The *TYR* gene had moderate to severe missense amino acid mutations in two AHM samples (p.P70S and p.M96_F98del), with a third sample having a CNV loss within the *OCA2* gene. In considering only the *TYR* point mutation/deletion, this represents 28.6% of the AHM compared with 3.0% of PM and 3.7% of other tumor samples ($P$ = 0.021 and 0.026

**Table 5. Somatic missense mutation or CNV change [a] in albinism genes within AHM patient tumor samples from the TCGA SKCM collection.**

| Patient | EE-A2GO | EE-A2GE | EE-A2GS | EE-A3J3 | BF-AAOU | EB-A4P0 | EB-A551 |
|---|---|---|---|---|---|---|---|
| **Gene/OCA** | | | | | | | |
| *TYR*/OCA1 | **P70S [b,c]** | 0 | **M96_F98del [b]** | 0 | 0 | 0 | 0 |
| OCA2/OCA2 | 0 | 0 | 0 | 0 | 0 | 0 | **Loss** |
| *TYRP1*/OCA3 | 0 | **S137N** | 0 | 0 | 0 | 0 | 0 |
| *SLC45A2*/OCA4 | 0 | 0 | 0 | 0 | 0 | 0 | 0 |
| *SLC24A5*/OCA6 | 0 | 0 | **Loss** | 0 | 0 | 0 | **Loss** |
| *MC1R* | 0 | 0 | 0 | 0 | 0 | 0 | 0 |
| *ASIP* | 0 | 0 | 0 | 0 | 0 | 0 | 0 |
| *CLCN7* | 0 | 0 | 0 | 0 | 0 | 0 | 0 |

[a] Restricted to moderate to severe somatic missense mutation or CNV Loss

[b] *TYR* point mutations seen in 2/7 AHM (28.6%), 13/431 PM (3.0%) and 364/9732 other cancers (3.7%), where genotypes are available. AHM vs PM $P$ = 0.021; AHM vs other cancers $P$ = 0.026

[c] rs372689330

respectively). No mutations were seen in the *MC1R* or *ASIP* pigmentation genes nor the non-pigment related *CLCN7* gene.

## Discussion

These results clearly demonstrate that rare albinism-associated variants in *TYR*, and the *OCA2* p.V443I variant, are more frequent in individuals with melanoma as compared to controls. This is in keeping with the recent report, by Goldstein and colleagues [51] that familial melanoma cases in the USA had an increased burden of rare germline variants in *TYR* and *OCA2*, and sporadic melanoma cases an increased frequency of mutations in *TYR*. In particular, the *TYR* stopgain p.R402* mutation (rs62645917) co-segregated with melanoma in five familial cases and has a population frequency of ~$3x10^{-5}$. In the present collection, rare variant burden tests showed that familial and population-based cutaneous melanoma patients tended to have higher frequencies of rare germline variants in albinism genes including *TYR* and *OCA2*. Of further significance, our study demonstrated that rare variants of *TYR* and the *OCA2* p.V443I allele were even more frequent in individuals with AHM as compared to those with PM.

These results suggested that rare heterozygous variants in pigmentation genes may play a role in melanoma susceptibility, probably through partially impaired melanogenesis. This is highly plausible given that carrying an *OCA2* albinism allele was associated with lighter skin and eye color than seen in the generally darker pigmented population in a Polynesian community [52]. In another example demonstrating skin lightening in a Japanese study [53], the *OCA2* p.A481T polymorphism was strongly associated with melanin index, correlating with lighter skin color ($P = 6.18x10^{-8}$), and is an albinism allele recognized by Lasseaux et al [3]. It is therefore reasonable to assume that heterozygosity for deleterious mutations at albinism loci in populations with European ancestry would also result in lighter skin color, but this may not be as obvious given the light skin type of most Europeans. As lighter skin color is recognized as a risk factor for melanoma [54] this would provide a plausible explanation for the raised frequency of *TYR* and *OCA2* p.V443I albinism/deleterious gene allele carriers in melanoma cases.

The focus of this analysis was on rare alleles at below 1% frequency, with many of these being assigned a deleterious status based on the association with albinism [3, 43] (Albinism database http://www.ifpcs.org/albinism/index.html). Given that they are rare, it is unsurprising that only one of the *TYR* alleles, p.T373K (S1 Table, $P = 0.02$ for AHM), overlapped with those reported by Goldstein [51]. The *OCA2* p.V443I was not reported upon by Goldstein et al., as it occurred at greater frequency than 0.1% in their collection which was selected as the cut off frequency in this earlier study, however the *OCA2* p.C777Y and p.Y342C alleles are reported on here (seen only in controls, see S1 Table).

Although melanoma in albinism patients is rare, it is almost invariably presents as amelanotic melanoma as seen in *TYR*-mutated OCA1 patients [55]. Deleterious variants in *TYR*/OCA1 were more common in AHM melanoma cases than in PM cases and the same was true for the *OCA2* hypomorphic allele p.V443I. In our study, over half of the AHM patients had multiple primary melanoma (MPM) [18], and in all of these cases at least one was a PM. Furthermore, although 33 of the AHM patients had fair skin, 12 AHM patients had moderate skin complexion (5 single and 7 MPM). Therefore, AHM does arise on constitutively pigmented skin types, and tumor color can vary within the same patient. As such, we postulate that a loss of melanin in melanoma tumor tissue in AHM may occur due to loss of heterozygosity (LOH) or somatic mutation of the functional copy of the *TYR* or *OCA2* genes, with the remaining deleterious albinism allele(s) leading to a deficiency in melanogenesis or the observed lack of pigment.

Examination of seven AHM samples from the TCGA SKCM collection did not reveal any deleterious of albinism alleles in the germline of these patients, as such whether this genetic loss of function mechanism actually occurs in AHM melanoma tissue remains to be experimentally confirmed. However, data supportive of this hypothesis is that the matched tumor tissues revealed that somatic mutations or CNV changes were present in albinism genes. Moreover, the rate of *TYR* somatic mutation was found to be significantly higher in AHM compared with PM or other cancers. Chromosomal deletion and LOH is a common event in melanoma pathogenesis [23, 56], so it seems plausible that in some tumors, hypopigmentation could arise in heterozygote carriers of albinism variants due to LOH.

Common noncoding polymorphisms in the *HERC2/OCA2* locus have previously been associated with melanoma [57, 58], with both *HERC2* rs12913832 and *OCA2* intron haplotype region SNPs associated with eye color and melanoma [59–63]. The nonsynonymous p.V443I hypomorphic allele [64] was first reported in two melanoma cases [57], then identified in an *OCA2* albinism/familial melanoma family as a compound heterozygote with the deleterious p. L734R allele [43]. In co-segregation analysis of the extended pedigree, both of these *OCA2* deleterious alleles were analyzed as melanoma risk variants, with an OR of 6.55. In a recent report of a multi-gene panel screening of Dutch non-*CDKN2A/CDK4* melanoma families, 9 rare pathogenic variants of *OCA2* were found [65]. The p.V443I and p.N489D alleles were detected at double the frequency of Dutch GoNL reference database controls (1.8% and 0.71%) similar to the frequency of these alleles in all melanoma cases reported here (1.81% and 0.09%, S1 Table). However, the frequency of the p.N489D allele did not statistically differ according to disease status in the BNMS cohort. Notably, a p.V443I homozygous Dutch patient was reported as having 3 primary melanomas, as was another individual who was biallelic for *OCA2* [65]. In another recent study of Australian and Danish melanoma families [66], the *OCA2* p.V443I frequency was increased in cases (6 carriers in 107 cases, crude allele frequency 2.8%) and the *OCA2* p.N489D was detected in three cases from one pedigree (1.4%). This is comparable to our data, where 7 of 106 (3.3%) of multiple primary melanoma cases had a report of an affected first degree relative and p.N489D was present in one multiple primary melanoma patient and five (BNMS and MGRB) controls. If one combines these two study results for the p.N489D allele (4/676 cases, 0.3%; 5/1806 controls, 0.1%), this still does not quite reach statistical significance.

A surprising finding was the significant association of the extended promoter haplotype in the first intron of *OCA2* with AHM (Table 4). The rs7174027*A allele was absent in all 45 AHM samples analyzed, but present in 7.1% of pigmented cases and 9.43% of BNMS controls, giving the highest *P*-value in comparing SNPs between AHM and PM (0.0005) and any BNMS cases vs controls ($8x10^{-5}$). The intronic haplotype tagged by rs7174027 acts independently of the most highly associated SNP for eye color, rs12913832, in its effects on AHM. It was reported earlier that the three-SNP haplotype based on rs7495174*A/G, rs4778241*C/A, and rs4778138*A/G (termed blue-eye associated haplotype 1, BEH1 [45]) was a recessive modifier associated with lighter pigmentary phenotypes [12]. This *OCA2* haplotype block, including rs7174027, could potentially lead to a reduction in expression of *OCA2* levels so as to functionally contribute to the hypopigmented phenotype in the AHM cohort, although direct experimental testing has not yet found any evidence for this [67]. However, in support of a transcriptional model for intron 1 regulating *OCA2* expression, a melanocyte-specific eQTL analysis of human primary melanocyte cultures reported that the rs4778138*A/G SNP (in the BEH1 haplotype block, Table 4) significantly affected expression ($P = 8.72x10^{-8}$ and slope of rank normalized expression of 0.62), where individuals homozygous for the A allele had lower expression than the G homozygotes [46]. The rs4778138*G SNP has previously been identified as the protective allele for melanoma [62].

Our study makes a significant contribution to the literature in expanding the understanding of how albinism genes influence risk for both amelanotic and pigmented melanomas. Populations with European ancestry are broadly recognized for having a higher risk for melanoma than those of non-European ancestry. Therefore, it is imperative to identify those genetically at-risk individuals within this group to optimize clinical surveillance and thus potentially improve patient outcomes. The main limitation to this study is the smaller AHM sample size which constrained the studies power to detect novel genes, but this candidate gene approach provides a solid foundation to conduct prospective future studies further exploring the OCA albinism genes in melanoma subtypes.

## Conclusion

It has been widely discussed how the dermatological and ophthalmological phenotypic heterogeneity in albinism impedes the establishment of phenotype–genotype correlations [3]. AHMs occur more frequently in those with fair hair and lighter skin, but hypopigmented melanomas can also occur in those with brown or black hair and, therefore, the commonly utilized fair hair/light skin phenotype profile risks missing the latter group. Individuals with darker hair may still be heterozygous carriers of deleterious *TYR* and *OCA2* p.V443I albinism genes that will have effects on melanogenesis in the skin and so increase the risk of these hypopigmented melanomas. Individuals carrying OCA deleterious gene variants have an increased risk of CM development, which might be more likely to present as AHM, due to a second hit to the gene during tumor development. Indeed, in the TCGA SKCM collection a higher incidence of somatic mutation of the *TYR* gene was seen in AHM compared with PM tumor samples. This work also contributes significantly to the literature by expanding a region of interest in the first intron of *OCA2* to describe a haplotype with a significant difference in the allele frequency between our AHM and PM cohorts. Most significantly, the rs7174027*A SNP in the first intron of the *OCA2* gene, which may influence the promoter activity of this region, was absent in AHM patients. This *OCA2* haplotype may lead to a change in expression of *OCA2* levels, functionally contributing to the hypopigmented tissue in the AHM cohort.

## Supporting information

**S1 Fig. Brisbane Naevus Morphology Study flowchart of control, pigmented melanoma and amelanotic/hypomelanotic melanoma participants.** AHM amelanotic/hypomelanotic melanoma; WES whole exome sequencing.
(TIF)

**S1 Table. *TYR* and *OCA2* gene alleles in pigmented melanoma and amelanotic/hypomelanotic melanoma patients.**
(DOCX)

**S2 Table. *OCA3, OCA4, OCA5, OCA6, OCA7* gene alleles in pigmented melanoma and amelanotic/hypomelanotic melanoma patients.**
(DOCX)

**S3 Table. Pigmentation gene alleles in pigmented melanoma and amelanotic/hypomelanotic melanoma patients.**
(DOCX)

**S4 Table. *MC1R* genotype and allele frequencies in PM and AHM from the TCGA SKCM collection.**
(DOCX)

**S1 File. Variant call file of rare variants of albinism and pigmentation genes with ≤1% frequency in gnomAD [39] seen in 45 AHM patients from the BNMS.** The genes reported in this study include *TYR, OCA2, TYRP1, SLC45A2, SLC24A5, LRMDA, KITLG, POMC, SLC24A4, TPCN2.*
(VCF)

**S2 File. Variant call file of rare variants of albinism and pigmentation genes with ≤1% frequency in gnomAD [39] seen in 389 PM patients from the BNMS.** The genes reported in this study include *TYR, OCA2, TYRP1, SLC45A2, SLC24A5, LRMDA, KITLG, POMC, SLC24A4, TPCN2.*
(VCF)

**S3 File. Variant call file of rare variants of albinism and pigmentation genes with ≤1% frequency in gnomAD [39] seen in 652 unaffected controls from the BNMS.** The genes reported in this study include *TYR, OCA2, TYRP1, SLC45A2, SLC24A5, LRMDA, KITLG, POMC, SLC24A4, TPCN2.*
(VCF)

**S4 File. ANNOVAR annotated variants (53) of albinism and pigmentation genes with ≤1% frequency in ExAC database [39] seen in 383 melanoma cases from the BNMS subject to Whole Exome Sequencing.** The genes reported in this study include *TYR, OCA2, TYRP1, SLC45A2, SLC24A5, LRMDA, KITLG, POMC, SLC24A4, TPCN2.* Common alleles indicated in S1 Table to S3 Table are also included.
(XLSX)

**S5 File. ANNOVAR annotated variants of unknown significance for *TYR* and *OCA2* genes.** The *TYR* and *OCA2* gene variants listed in S1 Table as deleterious by *in silico* analysis using Polyphen2 [30] or MutationTaster [32].
(XLSX)

## Acknowledgments

The results published here are in whole or part based upon data generated by the MRGB Partners: https://sgc.garvan.org.au/initiatives/mgrb.

## Author Contributions

**Conceptualization:** David L. Duffy, Brian De'Ambrosis, B. Mark Smithers, Helmut Schaider, H. Peter Soyer, Richard A. Sturm.

**Data curation:** Jenna E. Rayner, David L. Duffy, Kasturee Jagirdar, Katie J. Lee, Erin K. McMeniman, Aideen M. McInerney-Leo.

**Formal analysis:** Jenna E. Rayner, David L. Duffy, Darren J. Smit, Kasturee Jagirdar, Erin K. McMeniman, Aideen M. McInerney-Leo, Mitchell S. Stark.

**Funding acquisition:** Helmut Schaider, H. Peter Soyer, Richard A. Sturm.

**Investigation:** Jenna E. Rayner, David L. Duffy, Darren J. Smit, Kasturee Jagirdar, Katie J. Lee, Erin K. McMeniman, Aideen M. McInerney-Leo.

**Methodology:** David L. Duffy, Aideen M. McInerney-Leo, Helmut Schaider, H. Peter Soyer, Richard A. Sturm.

**Project administration:** H. Peter Soyer, Richard A. Sturm.

**Resources:** Brian De'Ambrosis, B. Mark Smithers, Erin K. McMeniman, Helmut Schaider, H. Peter Soyer.

**Software:** David L. Duffy.

**Supervision:** H. Peter Soyer, Richard A. Sturm.

**Validation:** Jenna E. Rayner, David L. Duffy.

**Visualization:** David L. Duffy.

**Writing – original draft:** Jenna E. Rayner, David L. Duffy, Richard A. Sturm.

**Writing – review & editing:** Darren J. Smit, Kasturee Jagirdar, Katie J. Lee, Brian De'Ambrosis, B. Mark Smithers, Erin K. McMeniman, Aideen M. McInerney-Leo, Helmut Schaider, Mitchell S. Stark, H. Peter Soyer.

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
