## [Decision Letter · Decision Letter 0]

21 Apr 2020

PONE-D-20-07313

Albinism variants in individuals with amelanotic/hypomelanotic melanoma: increased carriage of TYR and OCA2 variants and their somatic mutation in AHM tumor tissue

PLOS ONE

Dear Dr. Sturm,

Thank you for submitting your manuscript to PLOS ONE. After careful consideration, we feel that it has merit but does not fully meet PLOS ONE’s publication criteria as it currently stands. Therefore, we invite you to submit a revised version of the manuscript that addresses the points raised during the review process.

Please address the points brought up by the reviewers. Additionally, all the tables, including supplementary, have to clearly indicate the statistical tests used. There is lot of information provided in footnotes, but not on statistical analyses. The point of multiple testing was brought up in the previous review but not addressed. If you suggest that SMMAT takes care of this issue, it has to be explained. 

Supplementary materials have track changes and highlight marks that need to be removed. 

We would appreciate receiving your revised manuscript by Jun 05 2020 11:59PM. To enhance the reproducibility of your results, we recommend that if applicable you deposit your laboratory protocols in protocols.io, where a protocol can be assigned its own identifier (DOI) such that it can be cited independently in the future. For instructions see: http://journals.plos.org/plosone/s/submission-guidelines#loc-laboratory-protocols

We look forward to receiving your revised manuscript.

Kind regards,

Ludmila Prokunina-Olsson, PhD

Academic Editor

PLOS ONE

Journal Requirements:

2. Please provide additional details regarding participant consent. In the ethics statement in the Methods and online submission information, please ensure that you have specified whether consent was informed.  If your study included minors, state whether you obtained consent from parents or guardians.

3. Thank you for stating the following in the Competing Interests section: "I have read the journal's policy and the authors of this manuscript have the following competing interests: HPS is a shareholder of MoleMap NZ Limited and e-derm consult GmbH, and undertakes regular teledermatological reporting for both companies. HPS is a Medical Consultant for Canfield Scientific Inc. and MetaOptima Technology Inc., a Medical Advisor for First Derm, and has a Medical Advisory Board Appointment with MoleMap NZ Limited. The other authors have no conflicts of interest to declare."

Reviewers' comments:

Reviewer's Responses to Questions

**Comments to the Author**

1. Is the manuscript technically sound, and do the data support the conclusions?

Reviewer #1: Yes

Reviewer #2: Partly

2. Has the statistical analysis been performed appropriately and rigorously? 

Reviewer #1: Yes

Reviewer #2: I Don't Know

3. Have the authors made all data underlying the findings in their manuscript fully available?

Reviewer #1: Yes

Reviewer #2: Yes

4. Is the manuscript presented in an intelligible fashion and written in standard English?

Reviewer #1: No

Reviewer #2: No

5. Review Comments to the Author

Reviewer #1: In the research article entitled “Albinism variants in individuals with amelanotic/hypomelanotic melanoma: increased carriage of TYR and OCA2 variants and their somatic mutation in AHM tumor tissue”, the authors use whole exome sequencing to look for rare variants (<1% MAF) in albinism-associated genes in a small group of patients who had amelanotic or hypomelanotic melanoma. They show that rare variants in TYR (OCA1) are more common in AHM than PM patients. No rare variants were found in OCA2,3,4,6,7 in the AHM patients but they did find higher incidence of a variant (p.V443I) in OCA2 that is below <1% in gnomAD but not in their own control group. They then analyzed blue eye color-associated polymorphisms at OCA2/HERC2 and found a significant difference in rs7174027*A (associated with darker eye color) between AHM (not detected) and PM (also controls). Lastly, the authors suggest that AHM could occur due to LOH or somatic mutation of the functional copy of TYR or OCA2. The authors support this model using TCGA melanomas which found higher incidence of mutations or CNV in albinism genes in amelanotic melanomas (n=7) compared to pigmented melanomas (n=431).

The authors have sufficiently responded to and corrected for previous reviewer’s comments.

Minor Issues

There were 52 unknown pigment status melanomas sequenced as described in the text (line 182-185) and Figure S1 which were then not mentioned for the rest of the paper. Please remove this or clarify if and how these melanomas were used in this article.

Due to the low number of amelanotic melanomas analyzed from TCGA melanoma dataset, the conclusions drawn from these seven tumors, while intriguing, are suggestive and inclusion of “and their somatic mutation in AHM tumor tissue” in the title could be misleading. Please consider revising your title to not over state these findings.

Small clarifications / fixes

Lines 221 to 223: Here the authors start with 18 variants and remove the two high frequency gnomAD variants which would leave 16 variants but the next sentence talks about the 15 rare variants. Would like a sentence explicitly stating that the p.V443I variant is common in your set in between these two sentences.

In Figure 1, please label the directionality of the OCA2 and HERC2 genes or label the haplotype region reference in line 306.

There are different fonts being used in Table 3.

There is a discrepancy on line 342, which states 463 PMs, and the footnote of Table 5, which states 431 PMs, for the TCGA analysis. Please resolve this discrepancy.

The results section starting at line 337 has multiple duplicate words throughout.

Reviewer #2: A manuscript by Rayner and co-authors entitled “Albinism variants in individuals with amelanotic/hypomelanotic melanoma: increased carriage of TYR and OCA2 variants and their somatic mutation in AHM tumor tissue” may potentially be of interest to PLOS One readers, but the way the authors presented their research and organized the data in the manuscript is very difficult to follow and understand.

In the RESULTS, Analysis of albinism gene alleles in PM and AHM cases section the authors write:

Three of these TYR rare variants

202 (p.A23T, p.T373K and p.P460L) were found in the AHM population in comparison to six

203 (p.R217Q, p.V275F, p.R299H, p.N371T, p.T373K, p.P460L) in the larger PM group. Overall,

204 rare TYR variants were identified in 4.67% of AHM cases compared to 1.76% of PM cases, and

205 1.14% of MGRB controls (Table 1).

It is unclear how the value “4.67% of AHM cases” was calculated. As the sentence is written, one may believe that the authors sequenced 28 AHM cases and found 3 rare TYR variants in the patients. However, 4.67% of 28 AHM cases would be (28/100)*4.67=1.3076. Based on Table1 (column5 X row3), one wonders if the value 4.67 is meant to be understood as MAF (Minor Allele Frequency). The way it’s calculated, however is not commonly understood as MAF.

Data in the tables are poorly organized and difficult to understand. For instance, in Table 1, the same column contains different types of variables: the intersection of column “MGRB Control; N (MAF%); Total =1144; WES” with row 2 “Total observed rare alleles TYR (<1%) assayed” shows a value of “12 of 18”; and the intersection of the same column with row 3 “Sum TYR (MAF%) observed rare alleles (<1%) assayed” shows a value of “26 (1.14)”.

It’s unclear why p-values for statistical tests are placed in the row called “Sum TYR (MAF%) observed rare alleles (<1%) assayed.”

It is not immediately clear what the difference is between “Total observed rare alleles TYR (<1%) assayed” and “Sum TYR (MAF%) observed rare alleles (<1%) assayed”. The table headers should be concise and clear and the table legend should explain the headers if they are not self-evident.

One observes similar issues with Table 2. The way the data is organized is difficult to follow, the column and row headers are poorly labeled and explained. In addition, there are cells that the authors forgot to fill in. For example, what are the Total N(%) values for TYR in “Controls versus AHM cases”?

Additionally, data presented in the table are inconsistent. For example: columns 2 and 7 “Total N(%)” should contain percent values in parentheses in every cell.

Table 3: references used in the footnotes, e.g. “Lek et al., 2016 [18]” and “Zhang et al., 2018 [46],” do not match the list of references. One observes the same problem with Supplementary Table S1. Omissions like these do not increase the reader’s confidence in the quality of the manuscript.

Overall, the manuscript is full of omissions, mistakes, poorly organized content and is not clearly written. With this low quality of presentation, it is difficult to evaluate the true scientific value and merits of the research.

6. PLOS authors have the option to publish the peer review history of their article (what does this mean?). If published, this will include your full peer review and any attached files.

Reviewer #1: No

Reviewer #2: No

---

## [Author Response · Author response to Decision Letter 0]

26 Jun 2020

The Editors

PLOS One

https://journals.plos.org/plosone

 Friday 26th June, 2020

Dear Editors:

We are pleased to hear that you are willing to consider a revised version of our manuscript no: PONE-D-20-07313 as an Original Article in PLOS One.

New Title: Germline and somatic albinism variants in amelanotic/hypomelanotic melanoma: increased carriage of TYR and OCA2 variants

Authors: Jenna E. Rayner, David L. Duffy, Darren J. Smit, Kasturee Jagirdar, Katie J. Lee, Brian De’Ambrosis, B. Mark Smithers, Erin K. McMeniman, Aideen M. McInerney-Leo, Helmut Schaider, Mitchell S. Stark, H. Peter Soyer and Richard A. Sturm

The revised manuscript has addressed the reviewers’ concerns and these changes have been added to the electronic version of “Revised Manuscript with Track Changes.docx”, and the responses described below. The text is now 4890 words, Figure 1 has been modified as have all Tables and Supplementary Tables. We apologise that the submission is being sent after 5th of June suggested for preparation of the revision.

This cover note is included as an attachment “Response to Reviewers.docx”. The manuscript with highlighted changes is uploaded as “Revised Manuscript with Track Changes.docx”. The unmarked version of the manuscript is uploaded as “Revised Manuscript Clean Copy.docx”.

Editorial Comments:

Please address the points brought up by the reviewers. Additionally, all the tables, including supplementary, have to clearly indicate the statistical tests used. There is lot of information provided in footnotes, but not on statistical analyses.

This is now done, in addition to the other changes described below.

Table 1 has removed the denoted statistical tests in the footnotes and these are now in the headings of the columns.

Table 2 has been modified to spell out Odds Ratio and Trend X2 (P-value).

Table 3 has the use of fonts corrected.

Table 4 has been modified to spell out Odds Ratio and define Pr(>Chi) in the footnotes. The significance is now defined for **, ***

Table 5 has been corrected.

Table S1, S2 and S3 now spell out X2 Statistical Tests (P-value) in the heading to the columns and Table S1 now explicit where the SMMAT test is used in the headings.

Table S4 has removed the track change highlights and references now in correct format.

The point of multiple testing was brought up in the previous review but not addressed. If you suggest that SMMAT takes care of this issue, it has to be explained. 

The original comment and rebuttal in our first cover letter was, 

>2. Multiple testing correction was not mentioned either for single-variant testing or >gene-based testing. 

>Although we present X2 tests for each variant and report the significance in the supplementary 

>Tables, the SMMAT test is more appropriate that allows for the number of variants per gene.

We have added these sentences to the Materials and methods section, under the heading Statistical analysis

“Allele frequency of each variant in each subgroup was compared using contingency X2 tests. All P-values are unadjusted for multiple testing, but should be interpreted as nested within the gene based tests.”

And two sentences below this,

“This correctly adjusts for the presence of multiple variants contributing to the test.”

In the results section we have modified these sentences,

“In considering the frequency of individual rare TYR variants in different subtypes of melanoma, the p.A23T variant showed the largest difference, occurring at a higher frequency in AHM vs PM (X2 unadjusted P=0.008).”

“The p.A23T and p.T373K variants were more common in AHM cases as compared to controls (X2 unadjusted P=0.006 and X2 P=0.02 respectively).”

“However, analyzing each variant separately revealed that the p.V443I variant occurred at higher frequency in AHM cases (4.44%) compared to PM cases (1.57%), BNMS controls (0.84%), or the MGRB (1.09%) (combined control comparison X2 P=.01), with comparison of any case nominally significant (X2 unadjusted P=0.04).”

“Most significantly, the rs7174027*A SNP flanking the OCA2 gene promoter region/transcription start site was absent in AHM patients, compared to a frequency of 7.1% in the PM group (X2 unadjusted P=0.0005) and 9.43% frequency in the BNMS control group (X2 unadjusted P=8x10-5).”

Supplementary materials have track changes and highlight marks that need to be removed. 

These are now removed in the revised Tables.

There are no changes to the authors financial disclosures.

Journal Requirements:

This is now done.

2. Please provide additional details regarding participant consent. In the ethics statement in the Methods and online submission information, please ensure that you have specified whether consent was informed. If your study included minors, state whether you obtained consent from parents or guardians.

In the Methods section this now reads,

“This study was approved by the Human Research Ethics Committee of Princess Alexandra Hospital (approval HREC/09/QPAH/162) and The University of Queensland (approval #2009001590) and conducted in accordance with the Declaration of Helsinki. Participants provided written informed consent and parental/guardian consent was also provided for those under 18 years.”

3. Thank you for stating the following in the Competing Interests section: "I have read the journal's policy and the authors of this manuscript have the following competing interests: HPS is a shareholder of MoleMap NZ Limited and e-derm consult GmbH, and undertakes regular teledermatological reporting for both companies. HPS is a Medical Consultant for Canfield Scientific Inc. and MetaOptima Technology Inc., a Medical Advisor for First Derm, and has a Medical Advisory Board Appointment with MoleMap NZ Limited. The other authors have no conflicts of interest to declare."

This is now declared in the Competing Interests section.

"This does not alter our adherence to PLOS ONE policies on sharing data and materials.”

Now done.

Understood.

Review Comments to the Author

Reviewer #1: In the research article entitled “Albinism variants in individuals with amelanotic/hypomelanotic melanoma: increased carriage of TYR and OCA2 variants and their somatic mutation in AHM tumor tissue”, the authors use whole exome sequencing to look for rare variants (<1% MAF) in albinism-associated genes in a small group of patients who had amelanotic or hypomelanotic melanoma. They show that rare variants in TYR (OCA1) are more common in AHM than PM patients. No rare variants were found in OCA2,3,4,6,7 in the AHM patients but they did find higher incidence of a variant (p.V443I) in OCA2 that is below <1% in gnomAD but not in their own control group. They then analyzed blue eye color-associated polymorphisms at OCA2/HERC2 and found a significant difference in rs7174027*A (associated with darker eye color) between AHM (not detected) and PM (also controls). Lastly, the authors suggest that AHM could occur due to LOH or somatic mutation of the functional copy of TYR or OCA2. The authors support this model using TCGA melanomas which found higher incidence of mutations or CNV in albinism genes in amelanotic melanomas (n=7) compared to pigmented melanomas (n=431).

We thank the reviewer for understanding and succinctly summarising the paper.

The authors have sufficiently responded to and corrected for previous reviewer’s comments.

This modified manuscript has been through two cycles of revision before being considered by PLOS One. We thank the reviewer for considering that we have sufficiently responded to the PLOS Genetics reviews.

Minor Issues

There were 52 unknown pigment status melanomas sequenced as described in the text (line 182-185) and Figure S1 which were then not mentioned for the rest of the paper. Please remove this or clarify if and how these melanomas were used in this article.

Figure S1 is the flow diagram of the 1233 BNMS participants to show how they were parsed in the analysis performed for this study. There were 581 melanoma cases with 383 individuals for whom WES was obtained. The genotype data for the 52 melanoma samples of unknown pigmentation status was included in the analysis presented in the Tables i.e.

Table 1, and Table S1 to S3 have a sub heading under Melanoma Cases “Total Melanoma cases N (MAF%) Total =581(WES =383)”. 

Table 2 has the sub heading under Controls versus any melanoma cases, “BNMS Any Cases”

Note we have revised Table S2 to be the same format for the headings as Table 1, S1 and S3.

We do not feel it is required to restate that 52 of the total melanoma samples that WES were of unknown pigmentation phenotype, it is sufficient to say that they are included in the Total or any melanoma cases in the headings to the Tables.

Due to the low number of amelanotic melanomas analyzed from TCGA melanoma dataset, the conclusions drawn from these seven tumors, while intriguing, are suggestive and inclusion of “and their somatic mutation in AHM tumor tissue” in the title could be misleading. Please consider revising your title to not over state these findings.

We have changed the title to,

“Germline and somatic albinism variants in amelanotic/hypomelanotic melanoma: increased carriage of TYR and OCA2 variants”

Small clarifications / fixes

Lines 221 to 223: Here the authors start with 18 variants and remove the two high frequency gnomAD variants which would leave 16 variants but the next sentence talks about the 15 rare variants. Would like a sentence explicitly stating that the p.V443I variant is common in your set in between these two sentences.

This now reads,

“There were 18 coding variants in OCA2 (Table S1, gnomAD MAF), two of which, p.R305W (5.05%) and p.R419Q (6.5%), have been described previously as being common [10], with a third variant p.V443I at 1.09% in MGRB controls (Table 1 and S1). Seven of the remaining 15 rare variants (<1%) have been reported as causative for OCA2 albinism …”

In Figure 1, please label the directionality of the OCA2 and HERC2 genes or label the haplotype region reference in line 306.

The direction of gene transcription is now indicated above the gene names in Figure 1.

There are different fonts being used in Table 3.

This is now corrected

There is a discrepancy on line 342, which states 463 PMs, and the footnote of Table 5, which states 431 PMs, for the TCGA analysis. Please resolve this discrepancy.

The issue is that only 431 of the genotypes are available for the 463 PM samples. The footnote has been changed to read,

“… in 2/7 AHM (28%), 13/431 PM (3.02%) and 364/9732 other cancers (3.7%), where genotypes are available. …”

The results section starting at line 337 has multiple duplicate words throughout.

The use of a number of duplicate words has been reduced.

Reviewer #2: A manuscript by Rayner and co-authors entitled “Albinism variants in individuals with amelanotic/hypomelanotic melanoma: increased carriage of TYR and OCA2 variants and their somatic mutation in AHM tumor tissue” may potentially be of interest to PLOS One readers, but the way the authors presented their research and organized the data in the manuscript is very difficult to follow and understand.

We appreciate that there is difficulty in following the text, this is bought on by using both Illumina and WES genotyping methods to report variants in our collection. This requires some explanation for how the numbers of variants are detected and then analysed using the cut off of 1% variant allele frequency. Notably the OCA2 V443I allele is slightly above this cut off and was analysed separately (see comment and revision in response to reviewer 1).

In the RESULTS, Analysis of albinism gene alleles in PM and AHM cases section the authors write:

Three of these TYR rare variants

202 (p.A23T, p.T373K and p.P460L) were found in the AHM population in comparison to six

203 (p.R217Q, p.V275F, p.R299H, p.N371T, p.T373K, p.P460L) in the larger PM group. Overall,

204 rare TYR variants were identified in 4.67% of AHM cases compared to 1.76% of PM cases, and

205 1.14% of MGRB controls (Table 1).

It is unclear how the value “4.67% of AHM cases” was calculated. As the sentence is written, one may believe that the authors sequenced 28 AHM cases and found 3 rare TYR variants in the patients. However, 4.67% of 28 AHM cases would be (28/100)*4.67=1.3076. Based on Table1 (column5 X row3), one wonders if the value 4.67 is meant to be understood as MAF (Minor Allele Frequency). The way it’s calculated, however is not commonly understood as MAF.

In the original Table 1 the row title was “Sum TYR (MAF%) observed rare alleles (<1%) assayed”

and the equivalent original Table S1 row title was “Sum (MAF% s) observed rare alleles OCA1/TYR (<1%)”, this indicated Minor Allele frequency, not as the reviewer has calculated it.

s Genotypes calculated as “missing completely at random” with respect to disease status and % then appropriately weighted for each denominator of the variants being summed.

We apologise for the miswording in the text. This is now changed to read,

“Overall, rare TYR variants were at a MAF of 4.67% in AHM cases compared to 1.76% in PM cases, and 1.14% in MGRB controls (Table 1 and Table S1).”

Data in the tables are poorly organized and difficult to understand. For instance, in Table 1, the same column contains different types of variables: the intersection of column “MGRB Control; N (MAF%); Total =1144; WES” with row 2 “Total observed rare alleles TYR (<1%) assayed” shows a value of “12 of 18”; and the intersection of the same column with row 3 “Sum TYR (MAF%) observed rare alleles (<1%) assayed” shows a value of “26 (1.14)”.

We have tried to clarify this changing the title of row 2 to, 

“Allelic spectrum for TYR (MAF<1%) in each subgroup”

The title of row 3 is now,

“Total number of TYR alleles (MAF(<1%)) observed in each subgroup”

These changes are now also made throughout the rest of the Tables for consistency.

It’s unclear why p-values for statistical tests are placed in the row called “Sum TYR (MAF%) observed rare alleles (<1%) assayed.”

The heading to the last three columns of this row was

Statistical Tests (P-value)

In the revised Table it now appears as

“SMMAT Tests (P-value)”

And below it is

“X2 Statistical Tests (P-value)”

As has been explained, this is now explicit in each of the Tables.

It is not immediately clear what the difference is between “Total observed rare alleles TYR (<1%) assayed” and “Sum TYR (MAF%) observed rare alleles (<1%) assayed”. The table headers should be concise and clear and the table legend should explain the headers if they are not self-evident.

Total observed rare alleles now becomes “Allelic spectrum …” as explained above

And

Sum TYR (MAF%) … becomes “Total number of …” as explained above and consistent for all tables.

One observes similar issues with Table 2. The way the data is organized is difficult to follow, the column and row headers are poorly labeled and explained. 

We have merged cells where appropriate to indicate which rows contrast the P-values we are testing.

In addition, there are cells that the authors forgot to fill in. For example, what are the Total N(%) values for TYR in “Controls versus AHM cases”?

These are now entered, apologies it was an oversight.

Additionally, data presented in the table are inconsistent. For example: columns 2 and 7 “Total N(%)” should contain percent values in parentheses in every cell.

We have removed the N (%) from the column titles of Table 2.

Table 3: references used in the footnotes, e.g. “Lek et al., 2016 [18]” and “Zhang et al., 2018 [46],” do not match the list of references. One observes the same problem with Supplementary Table S1. Omissions like these do not increase the reader’s confidence in the quality of the manuscript.

Now corrected. 

Overall, the manuscript is full of omissions, mistakes, poorly organized content and is not clearly written. With this low quality of presentation, it is difficult to evaluate the true scientific value and merits of the research.

The first reviewer was able to follow the text and appreciate the hypothesis and conclusions reached, but we agree it has not been easy to assimilate all the data in a clear manner. It would have been best to have WES done on all samples but this was not possible given the financial constraints and it was better to increase the genotype counts using the Illumina CoreExome data which was available for a superset.

As corresponding author I can be contacted at the following numbers and address:

Assoc. Prof Richard A. Sturm

Dermatology Research Centre

The University of Queensland Diamantina Institute (UQ-DI)

Level 5, Translational Research Institute (TRI) 

37 Kent Street, Woolloongabba, Brisbane, QLD 4102, Australia 

Tel.: 61 7 34437380

Email: r.sturm@uq.edu.au

I hope this report is found appropriate for re-review in PLOS One.

Sincerely yours,

Assoc. Prof. R.A. Sturm

---

## [Decision Letter · Decision Letter 1]

27 Jul 2020

PONE-D-20-07313R1

Germline and somatic albinism variants in amelanotic/hypomelanotic melanoma: increased carriage of TYR and OCA2

PLOS ONE

Dear Dr. Sturm,

Thank you for submitting your manuscript to PLOS ONE. After careful consideration, we feel that it has merit but does not fully meet PLOS ONE’s publication criteria as it currently stands. Therefore, we invite you to submit a revised version of the manuscript that addresses the points raised during the review process.

We look forward to receiving your revised manuscript.

Kind regards,

Ludmila Prokunina-Olsson, PhD

Academic Editor

PLOS ONE

Reviewers' comments:

Reviewer's Responses to Questions

**Comments to the Author**

1. If the authors have adequately addressed your comments raised in a previous round of review and you feel that this manuscript is now acceptable for publication, you may indicate that here to bypass the “Comments to the Author” section, enter your conflict of interest statement in the “Confidential to Editor” section, and submit your "Accept" recommendation.

Reviewer #1: All comments have been addressed

Reviewer #2: (No Response)

2. Is the manuscript technically sound, and do the data support the conclusions?

Reviewer #1: Yes

Reviewer #2: Partly

3. Has the statistical analysis been performed appropriately and rigorously? 

Reviewer #1: Yes

Reviewer #2: No

4. Have the authors made all data underlying the findings in their manuscript fully available?

Reviewer #1: Yes

Reviewer #2: Yes

5. Is the manuscript presented in an intelligible fashion and written in standard English?

Reviewer #1: Yes

Reviewer #2: Yes

6. Review Comments to the Author

Reviewer #1: (No Response)

Reviewer #2: The quality of the manuscript is somewhat improved after the revision; however, a number of issues remain:

1. A substantial number of missense variants of uncertain significance (in TYR and OCA2) were included in the analyses; however, their deleteriousness was not sufficiently documented/proven. For instance, TYR allele rs1373014646*G/A p.A23T e is one of the three variants (Supplementary Table S1) in the gene used by the authors to calculate the frequencies of deleterious alleles in cases and compare them to controls. However, this variant is a VUS and a more rigorous argument should be made justifying the inclusion of this variant into the analyses. The authors used individual tools (MutationTaster and PolyPhen) to assess deleteriousness of the variants in-silico, but this is not sufficient. Instead, they should use more advanced ensemble methods, (e.g. REVEL, CADD, MetaSVM, etc.). They should define the criteria for variant deleteriousness and include the scores for all variants in the main and supplementary tables. In addition, the authors should clearly identify the ClinVar status (P/LP) of the variants by inserting relevant columns into the tables rather than by using superscript labels.

2. Given the substantial number of statistical tests performed in the study, a multiple testing solution (e.g., FDR, Bonferroni correction, etc.) should be employed and the results should be included in the manuscript (text, Tables, Supplementary tables) alongside the nominal p-values.

3. Since no complete inactivation of TYR or OCA2, (e.g., both the first and the second hits in the genes), was observed in any of the tumor samples investigated, the authors’ statement in the Abstract’s last sentence: “We suggest that somatic loss of function at these loci could contribute to the loss of tumor pigmentation, consistent with this we found a higher rate of somatic mutation in TYR/OCA2 in amelanotic/hypomelanotic melanoma vs pigmented melanoma samples from The Cancer Genome Atlas Skin Cutaneous Melanoma collection,” remains highly speculative. This sentence should be removed from the Abstract.

4. Table 1 issues:

a) The data in the row named “Allelic spectrum for TYR (MAF<1%) in each subgroup“ does not look highly relevant and does not help the reader understand the data; it can be easily omitted from the Table. Also, consider replacing “(MAF<1%)” with “rare” and put it in the Table’s caption rather than keeping it in the row header.

b) Second row header shows “TYR/OCA1”. Italicize TYR only.

c) Fifth row header shows OCA2/OCA2. Reverse the order, italicize the first OCA2 (gene symbol) and keep the second OCA2 regular (disorder). Keep it consistent throughout the manuscript: there are numerous instances where gene and disorder names are switched.

d) In the columns named “Controls” and “Melanoma Cases”, “(MAF%)” is not Minor Allele Frequency, it is something else. Perhaps call it “Proportion of all variant TYR alleles observed in the subgroup.”

e) In the “Melanoma Cases” columns, “Total” and “WES” numbers are shown, but it’s not clear how the frequencies were calculated and whether “Total” or “WES” numbers were iused. For instance, for values in row 4: “12 (1.76)” and “3 (4.67)” it’s not clear what numbers were used as denominators to obtain the values shown in parentheses.

f) Rounding the numbers appears to be inconsistent. For instance, in row 7, 6(1.0) should appear instead of “6 (0.9).” There are other instances of this type of inaccuracy in the table.

5. Supplementary Table S1 issues:

a) It’s not clear what the difference is between “N/A” and “-“.

b) Both common allele MAFs and significant p-values are shown in bold font. These should be distinguished differently.

c) Similar to what has been mentioned above in regard to Table 1: In the columns named “Controls” and “Melanoma Cases”, “(MAF%)” is not Minor Allele Frequency for values shown in rows “Total number of TYR (or OCA2; or TYR and OCA2) alleles (MAF<1% s) observed in each subgroup.”

d) In the first column, consider leaving rs IDs only and moving the rest of the information to the third column.

e) Consider splitting the table into a series of smaller ones based on Gene/OCA. Also, consider putting the RefSeq IDs in the column header or table caption, instead of repeating them in every cell.

7. PLOS authors have the option to publish the peer review history of their article (what does this mean?). If published, this will include your full peer review and any attached files.

Reviewer #1: No

Reviewer #2: No

---

## [Author Response · Author response to Decision Letter 1]

17 Aug 2020

The Editors

PLOS One

https://journals.plos.org/plosone

 Monday 17th August, 2020

Dear Editors:

We are pleased to hear that you are willing to consider an additionally revised version of our manuscript no: PONE-D-20-07313R1 as an Original Article in PLOS One.

Title: Germline and somatic albinism variants in amelanotic/hypomelanotic melanoma: increased carriage of TYR and OCA2 variants

Authors: Jenna E. Rayner, David L. Duffy, Darren J. Smit, Kasturee Jagirdar, Katie J. Lee, Brian De’Ambrosis, B. Mark Smithers, Erin K. McMeniman, Aideen M. McInerney-Leo, Helmut Schaider, Mitchell S. Stark, H. Peter Soyer and Richard A. Sturm

The revised manuscript has addressed the second reviewers’ concerns and these changes have been added to the electronic version of Revised Manuscript with Track Changes.docx, and the responses described below. The text is now 4906 words, Tables 1, 3 and 5 have been modified as have all the Supplementary Tables and an additional Supplementary File 5 has been added. The submission is being sent before the 10th of September suggested for preparation of the revision.

This cover note is included as an attachment “Response to Reviewers.docx”. The manuscript with highlighted changes is uploaded as “Revised Manuscript with Track Changes.docx”. The unmarked version of the manuscript is uploaded as “Revised Manuscript Clean Copy.docx”.

There are no changes to the authors’ financial disclosures.

Two figure files have been uploaded to the PACE website. The files subsequently downloaded are submitted here:

Rayner Figure 1.tif

Rayner Figure S1.tif

The Material and Methods section details the approach we have used in the calling of variants and statistical analysis. We feel it is unnecessary for a separate protocol submission to be undertaken for such generally understood procedures. 

Response to Reviewers' comments:

Reviewer #1: All comments have been addressed

We thank the reviewer for considering that our revised manuscript had adequately addressed their concerns with the original submission.

Reviewer #2: The quality of the manuscript is somewhat improved after the revision; however, a number of issues remain:

We are pleased that the reviewer believes our revised manuscript has been improved by the revisions that were made. In response to the additional issues:

1. A substantial number of missense variants of uncertain significance (in TYR and OCA2) were included in the analyses; however, their deleteriousness was not sufficiently documented/proven. For instance, TYR allele rs1373014646*G/A p.A23T e is one of the three variants (Supplementary Table S1) in the gene used by the authors to calculate the frequencies of deleterious alleles in cases and compare them to controls. However, this variant is a VUS and a more rigorous argument should be made justifying the inclusion of this variant into the analyses. The authors used individual tools (MutationTaster and PolyPhen) to assess deleteriousness of the variants in-silico, but this is not sufficient. Instead, they should use more advanced ensemble methods, (e.g. REVEL, CADD, MetaSVM, etc.). They should define the criteria for variant deleteriousness and include the scores for all variants in the main and supplementary tables. 

We used footnotes in Table S1 to delineate the relationship of each allele to an albinism variant or predicted to be deleterious using in silico techniques that were given as Polyphen2 and Mutation Taster. However, our analysis was in fact much more extensive than this but we did not highlight this in the text.

e Deleterious by in silico analysis using Polyphen2 [30] or MutationTaster [32] 

f Pathogenic albinism variant reported by Lasseaux et al., 2018 [3]

g Albinism database http://www.ifpcs.org/albinism/oca1mut.html, albinism allele

h Clinical Significance assigned as pathogenic or likely pathogenic in NCBI ClinVar database

k Clinical Significance assigned as uncertain in NCBI ClinVar database

l Clinical Significance assigned as benign in NCBI ClinVar database

n gnomAD African population frequency 2.6%

o OCA2 albinism allele Hawkes et al., 2013 [43]

p Albinism database http://www.ifpcs.org/albinism/oca1mut.html, polymorphism

In analysing the potential deleterious nature of variants we used the ANNOVAR program and this output is already included. We did state in the text,

“A separate Excel file containing ANNOVAR annotated variants (53) of these 10 genes is also provided (Supplementary File 4).”

In this file the current annotation of the variants includes the following in silico prediction programs:

SIFT_score………NUMERIC VALUE

SIFT_pred…………….”deleterious/benign/etc

Polyphen2_HDIV_score

Polyphen2_HDIV_pred

Polyphen2_HVAR_score

Polyphen2_HVAR_pred

LRT_score

LRT_pred

MutationTaster_score

MutationTaster_pred

MutationAssessor_score

MutationAssessor_pred

FATHMM_score

FATHMM_pred

RadialSVM_score

RadialSVM_pred

LR_score

LR_pred

VEST3_score

CADD_raw

CADD_phred

GERP++_RS

phyloP46way_placental

phyloP100way_vertebrate

Using the File named S4_File.xlsx, one can sort and find the TYR p.A23T variant, the in silico analysis as listed above can then be seen in columns AA to AX.

The TYR p.A23T variant the reviewer highlights was found only once, notably in the AHM group (Table S1). I will list which variants were designated as deleterious only based on “e” for TYR. 

rs1373014646*G/A p.A23T e once in the AHM group, but also present in gnomAD

rs1160771486*A/G p.M179V e once in the MGRB control group, but also present in gnomAD

rs61754368*GT p.K243fs e once in the MGRB control group alone

rs200854796*C/T p.R298W e once in the BNMS melanoma group in total, but not in PM or AHM subgroups, also in present in gnomAD

rs543973275*TT p.T489fs e once in the BNMS melanoma group in total alone, but not in PM or AHM subgroups 

There was one example where the variant had other evidence besides that designated using “e”

rs200471520*G/A p.G154E e,g once in the MGRB control group, but also present in gnomAD

I will list which variants were designated as deleterious only based on “e” for OCA2.

rs147785669*T/A p.R76W e once in the MGRB control group, but also present in gnomAD

rs202126510*T/C p.N495D e once in the MGRB control group, but also present in gnomAD

rs755768280*G/A p.H615Y e twice in the MGRB control group, but also present in gnomAD

rs753088699*G/A p.A709V e once in the MGRB control group, but also present in gnomAD

There were three examples where the variant had other evidence besides that designated using “e”

rs34731820*A/G p.I370T e,g,k,p once in the PM group, but also present in gnomAD

rs768934658*A/C p.L734R e,k,o once in the PM group, but also present in gnomAD

rs776814755*C/T p.C777Y e,f,h once in the MGRB control group, but also present in gnomAD

Considering just the TYR p.A23T mutation, this occurs at a homologous position as the OCA3/TYRP1 p.A24T mutation listed in Table S2,

rs61758405*G/A p.A24T e,f

f Pathogenic albinism variant reported by Lasseaux et al., 2018 [3]

Both of these gene mutations are considered deleterious by MutationTaster_pred and the CADD scores are 14.48 and 14.23 respectively in the S4_File.xlsx. Although the TYR p.A23T mutation has not been clinically described as a deleterious allele, by homology to the TYRP1 protein p.A24T variant, which has been clinically associated with OCA3, we are confident that this will be a hypopigmenting allele of TYR. At least this argument more fully justifies its inclusion in the analysis here beyond the in silico analysis.

To simplify the in silico presentation we have now included the analysis of all the alleles indicated as “e” for TYR and OCA2 listed in Table S1 into a separate Excel file (Supplementary File 5; S5_File.xlsx). We are not able to give a lengthy argument about why each separate allele is included in the analysis in Table S1 and Table 1 (such as we have given for TYR p.A24T), but the results for the in silico analysis are all now made more clearly available in this newly added Excel file. Moreover, as suggested we also used the Ensembl web site to look at how their prediction tools (which substantially overlap the ones we have already used) assessed function. All the alleles we listed were considered deleterious as we indicated, a file showing this has been made but we feel it is unnecessary to add this to the paper at this stage.

The text on page 5 now reads, 

“A separate Excel file containing ANNOVAR annotated variants (53) of these 10 genes is also provided (Supplementary File 4), and specific in silico analysis of TYR and OCA2 variants of unknown significance are shown in Supplementary File 5.”

An additional legend has been added on page 34,

“S5 File. ANNOVAR annotated variants of unknow significance for TYR and OCA2 genes. The TYR and OCA2 gene variants listed in Table S1 as deleterious by in silico analysis using Polyphen2 [30] or MutationTaster [32].

At the time when considering which in silico prediction tools were best to choose from we looked at nine common MC1R gene variants that we have previously performed detailed cellular and biochemical functional analysis viz.,

Beaumont KA, Shekar SN, Newton RA, James MR, Stow JL, Duffy DL, Sturm RA.

Receptor function, dominant negative activity and phenotype correlations for MC1R variant alleles.

Hum Mol Genet. 2007 Sep 15;16(18):2249-60.

Beaumont KA, Newton RA, Smit DJ, Leonard JH, Stow JL, Sturm RA.

Altered cell surface expression of human MC1R variant receptor alleles associated with red hair and skin cancer risk.

Hum Mol Genet. 2005 Aug 1;14(15):2145-54

These variants have been analysed by in silico techniques by Hepp et al., 2015

Hepp D, Gonçalves GL, de Freitas TR.

Prediction of the damage-associated non-synonymous single nucleotide polymorphisms in the human MC1R gene.

PLoS One. 2015 Mar 20;10(3):e0121812. doi: 10.1371/journal.pone.0121812. eCollection 2015.

In comparing the functional and in silico analysis with our own pipeline analysis, we made the interpretation that the Polyphen2 and MutationTaster programs were the most consistent in the case of what we already knew about the genetic effects of MC1R variants in humans. We have now modified the footnotes in Tables S1 and S2 to highlight the additional analysis that was already performed, which was always present, but is now made obvious in Supplementary File 5.

“e Deleterious by in silico analysis using Polyphen2 [30] or MutationTaster [32], with all detailed prediction tools shown in Supplementary File 5”

In addition, the authors should clearly identify the ClinVar status (P/LP) of the variants by inserting relevant columns into the tables rather than by using superscript labels.

The Tables are already quite large, we see no reason to specifically remove these superscripts for ClinVar status and make then into a separate column,

h Clinical Significance assigned as pathogenic or likely pathogenic in NCBI ClinVar database

k Clinical Significance assigned as uncertain in NCBI ClinVar database

l Clinical Significance assigned as benign in NCBI ClinVar database

The ClinVar status is by no means as important for each variant as is the albinism allele data from Lasseaux which is currently indicated by the superscript,

f Pathogenic albinism variant reported by Lasseaux et al., 2018 [3]

These are supplementary files the reviewer is referring to and this data will not appear or change the presentation as it appears in a printed article. We wanted the headings of the Tables and Supplementary Tables to somewhat match and prefer not to make such changes to the Supplementary Tables at this stage.

2. Given the substantial number of statistical tests performed in the study, a multiple testing solution (e.g., FDR, Bonferroni correction, etc.) should be employed and the results should be included in the manuscript (text, Tables, Supplementary tables) alongside the nominal p-values.

As explained previously we made these changes in the methods section:

“Allele frequency of each variant in each subgroup was compared using contingency X2 tests. All P-values are unadjusted for multiple testing, but should be interpreted as nested within the gene based tests.”

And two sentences below this:

“This correctly adjusts for the presence of multiple variants contributing to the test.”

Thus, it is only the contingency X2 tests that can be considered for adjustment due to multiple testing. Throughout the text we have indicated that these are unadjusted viz.

In the results section we have modified these sentences:

“In considering the frequency of individual rare TYR variants in different subtypes of melanoma, the p.A23T variant showed the largest difference, occurring at a higher frequency in AHM vs PM (X2 unadjusted P=0.008).”

“The p.A23T and p.T373K variants were more common in AHM cases as compared to controls (X2 unadjusted P=0.006 and X2 P=0.02 respectively).”

“However, analyzing each variant separately revealed that the p.V443I variant occurred at higher frequency in AHM cases (4.44%) compared to PM cases (1.57%), BNMS controls (0.84%), or the MGRB (1.09%) (combined control comparison X2 P=.01), with comparison of any case nominally significant (X2 P=0.04).”

“Most significantly, the rs7174027*A SNP flanking the OCA2 gene promoter region/transcription start site was absent in AHM patients, compared to a frequency of 7.1% in the PM group (X2 unadjusted P=0.0005) and 9.43% frequency in the BNMS control group (X2 unadjusted P=8x10-5).”

To take into account a Bonferroni correction we have added this to the supplementary Tables.

Table 1, no change for statistical correction 

Table 2, no change

Table 3, this text has been added to the footnote,

“e Bonferroni corrected (two phenotypes x 6 variants) critical P=0.004 equivalent to a table wide �=0.05”

Table 4, no change

Table 5, no change for statistical correction

Table S1,

“u Bonferroni corrected (three phenotypes x 30 variants) critical P=0.0005 equivalent to a table wide �=0.05”

Table S2, 

“l Bonferroni corrected (three phenotypes x 33 variants) critical P=0.0005 equivalent to a table wide �=0.05”

Table S3, 

“f Bonferroni corrected (three phenotypes x 49 variants) critical P=0.00003 equivalent to a table wide �=0.05”

As the OCA2 p.V443I allele is a previously published variant in the literature we do not consider it appropriate to perform a correction seeing this is a preplanned sub analysis.

3. Since no complete inactivation of TYR or OCA2, (e.g., both the first and the second hits in the genes), was observed in any of the tumor samples investigated, the authors’ statement in the Abstract’s last sentence: “We suggest that somatic loss of function at these loci could contribute to the loss of tumor pigmentation, consistent with this we found a higher rate of somatic mutation in TYR/OCA2 in amelanotic/hypomelanotic melanoma vs pigmented melanoma samples from The Cancer Genome Atlas Skin Cutaneous Melanoma collection,” remains highly speculative. This sentence should be removed from the Abstract.

Although we did not observe a complete loss of TYR or OCA2 gene function in the TCGA sample set that was analysed, we did find a loss of each of these genes at a rate of 28% in AHM vs 3.02% in PM. This evidence supports our conjecture and we use the word “suggest” in the sentence that the assessor highlights. We have changed the sentence to read,

“We suggest that somatic loss of function at these loci could contribute to the loss of tumor pigmentation, consistent with this we found a higher rate of somatic mutation in TYR/OCA2 in amelanotic/hypomelanotic melanoma vs pigmented melanoma samples (28.6% vs 3.0%; P= 0.021) from The Cancer Genome Atlas Skin Cutaneous Melanoma collection.”

The TYR pP50S is a deleterious mutation in Ensembl and the rs number is now given as a footnote in Table 5, 

“c rs372689330”

4. Table 1 issues:

a) The data in the row named “Allelic spectrum for TYR (MAF<1%) in each subgroup“ does not look highly relevant and does not help the reader understand the data; it can be easily omitted from the Table. Also, consider replacing “(MAF<1%)” with “rare” and put it in the Table’s caption rather than keeping it in the row header.

Three rows from Table 1 describing the Allelic spectrum of alleles are now deleted. 

As stated above we tried to keep Table 1 somewhat consistent with Table S1, as such we would prefer to keep the more exact term MAF<1% rather than use “rare” in the title of the Table. The fact that the p.V443I frequency is above 1% is important in how we present the analysis and the reason for footnote b of the Table,

b Does not include OCA2 rs121918166*C/T p.V443I

b) Second row header shows “TYR/OCA1”. Italicize TYR only.

Corrected

c) Fifth row header shows OCA2/OCA2. Reverse the order, italicize the first OCA2 (gene symbol) and keep the second OCA2 regular (disorder). Keep it consistent throughout the manuscript: there are numerous instances where gene and disorder names are switched.

Corrected, with Table 5, Table S1 and Table S2 also now made consistent, and at certain points in the text.

d) In the columns named “Controls” and “Melanoma Cases”, “(MAF%)” is not Minor Allele Frequency, it is something else. Perhaps call it “Proportion of all variant TYR alleles observed in the subgroup.”

We will call it “combined MAF%” to allow it to fit into the Tables.

e) In the “Melanoma Cases” columns, “Total” and “WES” numbers are shown, but it’s not clear how the frequencies were calculated and whether “Total” or “WES” numbers were iused. For instance, for values in row 4: “12 (1.76)” and “3 (4.67)” it’s not clear what numbers were used as denominators to obtain the values shown in parentheses.

These numbers are carried over from Table S1. In reply to the previous comment from the reviewer we added this footnote,

s Genotypes calculated as “missing completely at random” with respect to disease status and % then appropriately weighted for each denominator of the variants being summed.

We do not wish to copy over unnecessary footnotes from Table S1 to Table 1.

f) Rounding the numbers appears to be inconsistent. For instance, in row 7, 6(1.0) should appear instead of “6 (0.9).” There are other instances of this type of inaccuracy in the table.

This row (OCA2) and the analogous row in Table S1 are now presented to 2 significant figures. This has also changed the TYR+OCA2 row totals.

5. Supplementary Table S1 issues:

a) It’s not clear what the difference is between “N/A” and “-“.

In Table S1 where the SNP was not on the Illumina Core Exome chip it is indicated by the “-”. 

The “N/A” is used in the genomAD column when the variant allele in not seen in that database.

We now give the definition of N/A in the footnotes.

b) Both common allele MAFs and significant p-values are shown in bold font. These should be distinguished differently.

Common allele when >1% MAF is now show in italics. Also in Table S2 and Table S3.

c) Similar to what has been mentioned above in regard to Table 1: In the columns named “Controls” and “Melanoma Cases”, “(MAF%)” is not Minor Allele Frequency for values shown in rows “Total number of TYR (or OCA2; or TYR and OCA2) alleles (MAF<1% s) observed in each subgroup.”

These rows have a separate heading to indicate the SMMAT Test was used.

The row names have been changes to “(combined MAF<1%)” again to fit into the Tables.

d) In the first column, consider leaving rs IDs only and moving the rest of the information to the third column.

This would make the Tables less readable and relatable to the text. The amino acid change is what is discussed throughout the text. The reader would find it more difficult to find the variant if it were placed in the third row. The third row indicates the position of the variant in the transcript.

No change.

e) Consider splitting the table into a series of smaller ones based on Gene/OCA. Also, consider putting the RefSeq IDs in the column header or table caption, instead of repeating them in every cell.

The Tables present the data economically and by relevance. The major conclusions from the paper come from the data presented as TYR/OCA1 and OCA2/OCA2 in Table S1 which then feeds into Table 1 which will appear in the printed manuscript. The data in Table S2 includes the other OCA genes which do not achieve statistical significance and Table S3 which looks at other examples of pigmentation genes for comparison – which also do not achieve statistical significance. I consider that the rs number and protein change go together well, they are distinguishable by being on separate rows. Any modification that creates more columns makes it harder to fit on the one page or results in a decrease in the size of the text.

No change.

As corresponding author I can be contacted at the following numbers and address:

Assoc. Prof Richard A. Sturm

Dermatology Research Centre

The University of Queensland Diamantina Institute (UQ-DI)

Level 5, Translational Research Institute (TRI) 

37 Kent Street, Woolloongabba, Brisbane, QLD 4102, Australia 

Tel.: 61 7 34437380

Email: r.sturm@uq.edu.au

I hope this revised report is found appropriate for publication in PLOS One.

Sincerely yours,

Assoc. Prof. R.A. Sturm

---

## [Editor Report · Decision Letter 2]

19 Aug 2020

Germline and somatic albinism variants in amelanotic/hypomelanotic melanoma: increased carriage of TYR and OCA2 variants

PONE-D-20-07313R2

Dear Dr. Sturm,

We’re pleased to inform you that your manuscript has been judged scientifically suitable for publication and will be formally accepted for publication once it meets all outstanding technical requirements.

Kind regards,

Ludmila Prokunina-Olsson, PhD

Academic Editor

PLOS ONE
---

## [Editor Report · Acceptance letter]

26 Aug 2020

PONE-D-20-07313R2 

Germline and somatic albinism variants in amelanotic/hypomelanotic melanoma: increased carriage of *TYR* and *OCA2* variants 

Dear Dr. Sturm:

I'm pleased to inform you that your manuscript has been deemed suitable for publication in PLOS ONE. Congratulations! Your manuscript is now with our production department. 

Kind regards, 

on behalf of

Dr. Ludmila Prokunina-Olsson 

Academic Editor

PLOS ONE